# Effect of Dietary Fiber on Reproductive Performance, Intestinal Microorganisms and Immunity of the Sow: A Review

**DOI:** 10.3390/microorganisms11092292

**Published:** 2023-09-12

**Authors:** Feng Qin, Wenyan Wei, Junjie Gao, Xuemei Jiang, Lianqiang Che, Zhengfeng Fang, Yan Lin, Bin Feng, Yong Zhuo, Lun Hua, Jianping Wang, Mengmeng Sun, De Wu, Shengyu Xu

**Affiliations:** 1Key Laboratory of Sichuan Province, Animal Nutrition Institute, Animal Disease-Resistance Nutrition, Ministry of Education, Ministry of Agriculture and Rural Affairs, Sichuan Agricultural University, Chengdu 611130, China; 2022214062@stu.sicau.edu.cn (F.Q.); 2022214007@stu.sicau.edu.cn (W.W.); gaojunjie@stu.sicau.edu.cn (J.G.); 71310@sicau.edu.cn (X.J.); che.lianqiang@sicau.edu.cn (L.C.); zfang@sicau.edu.cn (Z.F.); linyan@sicau.edu.cn (Y.L.); fengbin@sicau.edu.cn (B.F.); zhuoyong@sicau.edu.cn (Y.Z.); hualun@sicau.edu.cn (L.H.); wangjianping@sicau.edu.cn (J.W.); wude@sicau.edu.cn (D.W.); 2College of Science, Sichuan Agricultural University, Yucheng District, Ya’an 625014, China; sunmeng14391@163.com

**Keywords:** sow, dietary fiber, reproductive performance, gut microbiota, SFCAs, inflammatory factors

## Abstract

Dietary fiber is a substance that cannot be digested by endogenous digestive enzymes but can be digested by the cellulolytic enzymes produced by intestinal microorganisms. In the past, dietary fiber was considered an anti-nutrient component in diets because it could resist digestion by endogenous enzymes secreted by the intestine and has a negative effect on the digestion of energy-producing nutrients. However, due to its functional properties, potential health benefits to animals, and innate fermentability, it has attracted increasing attention in recent years. There are a plethora of studies on dietary fiber. Evidence suggests that dietary fiber can provide energy for pigs through intestinal microbial fermentation and improve sow welfare, reproductive performance, intestinal flora, and immunity. This is a brief overview of the composition and classification of dietary fiber, the mechanism of action and effects of dietary fiber on reproductive performance, intestinal microorganisms, and the immune index of the sow. This review also provides scientific guidance for the application of dietary fiber in sow production.

## 1. Introduction

The reproduction and growth of animals are affected by gene control and environmental factors. Nutrients in the environment can directly or indirectly affect gene expression and have an important impact on the life process. These nutrients include water, protein, lipids, minerals, carbohydrates, vitamins and dietary fiber. Almost all nutrients regulate gene expression. Its function is characterized by the fact that one nutrient can regulate the expression of a variety of genes, and gene expression is regulated by a variety of nutrients. Nutrients can not only affect the expression of genes related to cell proliferation, differentiation, growth and development, but also play an important role in regulating the expression of pathogenic genes.

Dietary fiber is now known as the “seventh largest nutrient” and is considered an important and sustainable source of nutrients. Adding appropriate amounts of fiber to their diet can improve animal welfare, increase their reproductive performance, litter size, live birth size and feed intake during lactation, and improve their fecal scores [1]. Under the conditions of restricted feeding during pregnancy, the utilization of dietary fiber by the sow is very low due to a lack of endogenous digestive enzymes for digesting dietary fiber. This can increase the satiety of the sow without adding excessive energy, thus reducing the abnormal behavior of the sow and increasing the number of offspring [2,3,4,5].

As for the fiber in the diet, it is mainly digested by the exogenous digestive enzymes produced by intestinal microorganisms. Therefore, with the addition of different amounts of fiber to the diet, the composition and diversity of the intestinal flora of the sow can be effectively changed. Their interaction also affects the intestinal flora of the sow, which can change its reproductive performance, intestinal health, intestinal microbial diversity, and immune processes [6]. When fiber is ingested, the gut microbiota changes to become more adapted to breaking down fiber, producing more short-chain fatty acids (SCFAs), which are involved in different immune functions. For example, butyric acid can be used as an energy source for colonic epithelial cells and participates in the repair of inflammatory bowel disease, while acetic acid and propionic acid are involved in the energy metabolism of liver [7]. Due to the beneficial effects of fiber in improving the welfare and reproductive performance of the sow, at present, there is a great deal of research on this subject. If the specific regulatory mechanisms underlying intestinal microorganisms and intestinal health and immunity can be explored and elucidated, the livestock industry would benefit.

## 2. Composition and Classification of Dietary Fiber

The definition of dietary fiber was not scientifically unified at the beginning. The term “dietary fiber” was first proposed by Hipsley to denote the undigestible components of plant cell walls. Dietary fiber was then formally defined by Trowell as “polysaccharide carbohydrates and lignin that resist digestion and absorption by mammalian digestive enzymes”. The main components of dietary fiber include cellulose, hemicellulose, lignin, pectin, fructus, oligosaccharides, resistant starch, and other substances, such as cellulose in the primary wall of plants and a small amount of pectin, hemicellulose and lignin; pectin in the spacer layer of the cell wall; cellulose, hemicellulose and lignin in the secondary cell wall; and so on [8,9]. 

Initially, the concept of crude fiber (CF) was introduced; CF was defined as consisting of the main components of plant cell walls, including hemicellulose, cellulose and lignin substances. However, this definition of CF was too general and simple and did not include all the fiber components in the feed. Hence, next, the concepts of neutral detergent fiber (NDF) and acid detergent fiber (ADF) were introduced. According to their properties, fibers are divided into NDF and ADF. ADF is defined as the part of plant cells that is insoluble in acid detergents, including cellulose, lignin and silicates. NDF is defined as the part of plant cells that is insoluble in neutral detergents, including hemicellulose, cellulose, lignin and silicates [10,11,12]. In addition, both types of CF, NDF and ADF, mainly represent the amount of insoluble fiber, and soluble fiber is lost more, so some scholars have proposed a classification based on solubility: soluble fiber (SF) and insoluble fiber (ISF). According to their solubility, fibers are divided into SF and ISF. SF refers to a kind of fiber that can be dissolved in water, absorb water, expand and be fermented by microorganisms in the large intestine, which is beneficial to probiotics. ISF is a type of fiber that cannot be dissolved in water and cannot be fermented by microorganisms in the large intestine [13,14]. 

There has been gradual progress in research on SF and ISF. SF has water-holding capacity and expansibility [15]. Due to its strong solubility in the gastrointestinal tract, SF can enhance the viscosity of the intestinal contents, affect their speed of movement and reduce the time taken for their exclusion from the body, thereby improving the digestibility of diets that are more easily fermented by the hindgut microorganisms to produce SCFAs and gases. In particular, butyrate can be used as an energy substance to supply energy to the host. On the other hand, SF enhances the viscosity of the intestinal contents and forms a mucous membrane, which hinders the mixing of digesters and food and reduces the rate of glucose absorption. ISF strengthens intestinal peristalsis, increasing satiety and fecal volume, which can accelerate the elimination of intestinal contents and reduce the retention time of intestinal toxins in the body [5,14,16]. 

Dietary fiber sources can be divided into plant sources, animal sources, microbial polysaccharides, seaweed polysaccharides and synthetic substances. The composition of dietary fiber is complex, and the chemical essence of fiber from different sources is quite different. There are differences in the relative content of cellulose, hemicellulose and other components, the glycosidic bond, polymerization degree and branch chain structure of molecules, which implies that fibers in the digestive tract have different functions in different animals.

## 3. Effects of Dietary Fiber on Sow Reproduction and Related Mechanisms

### 3.1. Effect of Dietary Fiber on Reproductive Performance of the Sow

Studies have found that dietary fiber has an important effect on the performance of the sow. Adding fiber to the diet of the sow can improve its reproductive ability and increase its total litter size, live litter size, birth weight and feed intake during lactation. One of the most important fibers is soluble feed fiber, which can be added to the diet to reduce intrauterine growth restriction (IUGR) and improve reproductive performance, both in the firstborn of the sow as well as subsequent reproductive performance [5,17,18]. 

Feng et al. showed that feeding diets containing 10.8%, 15.8% and 20.8% NDF given to sows of different parity from day 1 through day 90 of gestation could improve their reproductive performance. In parity 1, the total average number of piglets born per litter in the 10.8% NDF group was increased by 0.74 and 1.05 piglets over that in the 15.8% NDF group and the 20.8% NDF group, respectively; the average number of piglets born alive (healthy and weak) per litter was also increased by 1.01 piglets over that in the 20.8% NDF group. In parity 2, the average total number of piglets born per litter in the 15.8% NDF group was increased by 0.91 and 1.03 piglets over that in the 10.8% NDF group and the 20.8% NDF group, respectively; the average number of piglets born alive (healthy and weak) per litter was also increased by 0.92 and 0.95 piglets over that in the 10.8% NDF group and the 20.8% NDF group, respectively [19]. Che et al. pointed out in their study that dietary fiber can significantly increase the number of live piglets born in the sow. At the second parity, more piglets survived in the high fiber group, the weight of piglets was significantly increased, and the weight of piglets’ viscera was also significantly increased [5]. Ferguson et al. showed that sows were fed 6 diets (maintenance (M) diet, 1.8 × M, 2.6 × M or nutritionally balanced diets in which the content of fiber, protein or starch was increased) during the estrous cycle prior to insemination. The results showed that the embryo survival rate of the high-fiber diet group (88.20 ± 1.96%) was significantly higher than that of 1.8 × maintenance diet (81.25 ± 2.67%) and the other four groups; this would reduce IUGR and be beneficial to reproduction [20]. Loisel et al. showed that dietary fiber supplementation during late pregnancy affected colostrum composition (Lipids, IgA) but not colostrum production in sows and increased colostrum intake in low-birth-weight piglets. Preweaning mortality was lower in high fiber (6.2%) than low fiber (14.2%) litters. [4]. Diao et al. pointed out that the sows in the control group and the high fiber group were fed a basal diet and high fiber diet during pregnancy, respectively. The number of healthy litters of sows in the high fiber group was significantly increased, the litter birth weight of piglets in the high fiber group was significantly increased, and the number of live litters in the high fiber group tended to increase, but there were no significant effects on the total litter size, the number of live litters, and the individual birth weights [21]. Veum et al. showed that when the daily intake of basal gestation feed was equal between the two treatment groups, the group in which the diet was supplemented with 13.35% ground wheat straw had an increase in the litter size and total weight of piglets born and weaned as compared with the group in which the diet was not supplemented [1].

In conclusion, the reproductive performance of the sow can be greatly improved and enhanced by feeding it dietary fiber, but the optimal content level and source of dietary fiber are not accurately characterized. Each addition contains different amounts of soluble versus insoluble fiber, which may account for the difference in the reproductive performance of the sow. If the optimal ratio of soluble and insoluble fiber can be determined, it will further improve the reproductive performance of sows. Further in-depth research is therefore needed.

### 3.2. Mechanism by Which Dietary Fiber Improves Reproductive Performance of the Sow

The improvement in sow reproductive performance was mainly manifested as an increase in the total number of litters, the number of live litters, and the number of effective piglets born, and a reduced number of stillbirths. 

The improvement in oocyte quality and ovarian reserve is an important way to increase litter size, improve lifetime reproductive performance, and accrue economic benefits. Previous studies have shown that dietary fiber added to the diet of the sow will improve their intestinal function, ovarian reserve, effective follicle proportion, and reduce the atresia of antral follicles. Cao et al. pointed out that when gilts were fed a basic corn-soybean meal diet, the high-fiber group had a significantly increased length of bilateral uterine horn and the relative weight of the uterus, increased maturation of oocytes, an improved survival rate of embryos, and improved reproductive performance of gilts [22]. Alvarez et al. showed that when the sow were fed diets containing high amounts of lignin fiber (insoluble fiber, 15.8% of dietary dry matter) and lignin fiber (insoluble fiber, 4.9% of dietary dry matter), the latter increased the number of oocytes reaching the second meiotic metaphase of oocytes and improved the reproductive performance of the sow [23]. Cao et al. pointed out that follicle development and survival were sensitive to dietary fiber level. With the increase in dietary fiber feeding, the number of primary follicles and the total follicle number showed a linear increase, and the reproductive performance of the sow was improved [24].

Previous studies have shown that feeding sows a high-fiber diet improved follicle quality, oocyte maturation, and early embryo survival in gilts, which appeared to be associated with changes in the estradiol (E2) and luteinizing hormone (LH) profiles [20,25]. At the same time, some relevant studies have shown that feeding sows a high-fiber diet before mating could improve the quality of sow oocytes and the reproductive performance of primiparous sows, but this was not related to changes in LH plasma concentration or LH pulse frequency [26]. Previous studies have shown that follicle activation and survival were controlled by nutrient sensors AMP-activated protein kinase (AMPK) and mammalian target of rapamycin (mTOR), as well as apoptosis-related markers cysteine aspartic acid-specific protease 3 (Caspase-3) and the pro-apoptotic factor B-cell lymphoma 2 associated X protein (BAX). The first step in simultaneous folliculogenesis is the activation of primordial follicles, which is closely related to the cellular nutrient sensors AMPK or mTOR [27]. 

When animals consume fiber, they produce more SCFAs, such as acetic acid, propionic acid, and butyric acid, which can significantly increase the ratio of intracellular AMP:ADP, activate the AMPK signaling pathway, initiate the catabolic process in cells, inhibit the expression of pro-apoptotic factor Caspase-3 and Bax, enhance the resistance of cells, and improve the quality of follicles and enhance their development [22,24,28]. The phosphorylation of mTOR and its downstream target S6K in the ovary decreased linearly with increasing dietary fiber levels [24]. SCFAs can stimulate GPR41 and GPR43 to promote energy consumption and cause changes in the ratio of AMP:ADP in cells [29], thereby activating the AMKP signaling pathway in ovarian follicles. Activated AMPK can phosphorylate and activate the upstream signaling molecule tuberous sclerosis complex 2 (TSC-2) of the mTOR pathway and promote the formation of the TSC-1/TSC-2 complex [30,31]. TSC-1/TSC-2 can inhibit the activity of GTPase Ras-homolog enriched in brain (Rheb), the upstream protein of mTOR, downregulate the activity of mTOR, inhibit the phosphorylation of S6K, and inhibit the mRNA expression of *Hif1α* and *Vegfa* in its downstream genes, thereby negatively regulating protein synthesis, granulosa cell and oocyte growth, and ultimately inhibiting primordial follicle activation [32] (Figure 1). In turn, MAPK3/1 signaling, which is closely related to mTORC1 signaling in granulosa cells and involved in primordial follicle activation, is also inhibited, ultimately reducing primordial follicle activation and causing follicle bank depletion and premature ovarian failure [33]. Studies have shown that SCFAs produced by dietary fiber intake could act on intestinal chromaffin cells to upregulate the gene expression of tryptophan hydroxylase (TPH1) in chromaffin cells and promote serotonin synthesis (5-HT). Synthetic 5-HT reaches the ovarian tissue through the peripheral circulation to upregulate the expression of serotonin receptor genes and ultimately inhibit the activation of ovarian primordial follicles and reduce the expression of genes and proteins related to the apoptosis of follicle granulosa cells [22]. 

In conclusion, the reproductive performance of the sow can be improved after feeding it dietary fiber, which affects the maturity of sow follicles, follicle quality, embryo survival and reduces follicle atresia by decreasing the number of activated primordial follicles through the corresponding signaling pathways. 

## 4. Effects of Dietary Fiber on Intestinal Microflora of the Sow and the Underlying Mechanism

### 4.1. Effect of Dietary Fiber on the Gut Microbiota of the Sow

Gut microbiota is one of the very important ecosystems in the gut. The complex microbial community in the GI tract is composed of different microbial groups, including bacteria, archaea, ciliate and flagellate protozoa, anaerobic algal bacteria fungi and bacteriophages. Bacteria are the most abundant and studied microorganisms in this community [34].

Dietary fiber intake can significantly change the relative abundance and diversity of intestinal microorganisms as well as intestinal permeability. Dietary fiber is one of the important nutrients in sow diet. In addition to increasing the satiety of sows, after entering the intestine, fiber cannot be digested and absorbed because of the lack of endogenous fiber catabolic enzymes in the small intestine. Therefore, fiber often enters the large intestine and is used as a substrate by intestinal microorganisms, fermented and decomposed to produce bioactive metabolites, which changes the relative abundance and diversity of intestinal microorganisms. 

Liu et al. showed that the alfalfa intake of the sow during pregnancy increased the relative abundance of anti-inflammatory bacteria, including Firmicutes, Bacteroides and Proteobacteria, decreased the relative abundance of pro-inflammatory bacteria, significantly affected the relative abundance and diversity of intestinal microorganisms, and regulated the production of SCFAs in the intestine [35]. Heinritz et al. showed that a low-fat, high-fiber diet significantly increased the number of bifidobacteria in the cecum and colon and significantly decreased the number of Escherichia coli [36]. The sows that were fed a pea fiber diet significantly increased the number of Lactobacillus in the pig colon, while the sows that were fed a diet containing soy fiber increased the number of *E. coli* [37]. Yu, Miao et al. showed that a high-fiber diet (stevia rebaudiana residue) in pregnant sows significantly increased the relative abundance of g-Lachnospirace-XPB1014-group, g-Christensenellaceae-R-7-group and g-Ruminococcaceae-UCG-005 at the genus level and decreased the relative abundance of Treponema-2 [38]. Firmicutes and actinomycetes are the main reactive bacteria in the gut that catabolize dietary fiber. Dietary fiber is the main energy source for the gut microbiota, which means that the addition of appropriate dietary fiber can increase the abundance of specific microorganisms. Dietary fiber intake can alter the intestinal permeability of sows. Li et al. found that sows that were fed a high-fiber diet during pregnancy had a significant reduction in endotoxin in late-pregnancy plasma compared with sows fed a low-fiber diet [39]. Liu et al. found that alfalfa intake in pregnant sows significantly reduced serum reactive oxygen species (ROS) and endotoxin levels [35]. In addition, fiber intake during pregnancy could improve the expression of tight junction proteins in the intestine of offspring piglets [40]. 

There is a close relationship between dietary fiber and gut microbiota. Most bacteria that degrade soluble fiber are beneficial and ferment soluble fiber to gases and short-chain fatty acids. The source of dietary fiber has an important influence on the proportion of SCFAs produced [41]. Dietary fiber may have a positive or negative effect on animal health, depending on the source of the fiber [17,18,26]. Shang et al. found that dietary beet residue supplementation significantly increased the fecal acetate, butyrate and total SCFA concentrations in pregnant sows [40]. In addition, the differences in dietary fiber sources and gut microbial sensitivity partially determine the complexity of the gut microbial system and affect the relative richness and diversity of gut microbes [42,43,44]. 

In summary, fiber can change the relative richness and diversity of intestinal microorganisms and the composition of SCFAs after entering the gut, thereby regulating other physiological activities, providing more energy to the body and inhibiting the growth and development of harmful bacteria to protect intestinal health.

### 4.2. Mechanism by Which Dietary Fiber Influences the Intestinal Microflora in the Sow

In animals, there are a large number of intestinal microorganisms. For fiber-sensitive bacteria, dietary fiber is the main energy source, which can be broken down to produce SCFAs and gases [45]. There is a homeostatic balance mechanism in the intestinal microbial system. When certain substances increase to a certain concentration, they begin to inhibit the proliferation of unsuitable bacteria. At the same time, some bacteria adapt to this environmental change and proliferate rapidly. With the continuous intake of dietary fiber, the gut microbiome as a whole will reach a new homeostasis, which can more easily decompose and absorb nutrients. The intake of dietary fiber from different sources by the sow can have different effects on microbial diversity and abundance [41]. SCFAs are produced by the microbial fermentation of dietary fiber in the large intestine. Because the SCFAs produced are acidic, they will reduce the pH in the intestine and provide suitable growth conditions for probiotics in the intestine, such as Bifidobacterium. At the same time, they can inhibit the proliferation of acid-sensitive bacteria in the intestine, such as Escherichia coli acid-sensitive strains and other bacteria. Furthermore, they change the relative abundance and diversity of intestinal microbes, increase the proportion of beneficial bacteria, and reduce the proportion of pathogenic bacteria and harmful metabolites in the gut [36,37,42,43,44]. 

When animals consume insufficient dietary fiber, the SCFAs produced by microbial fermentation in the large intestine are reduced, which increases the intestinal pH, limits the growth of probiotics and eventually alters the relative abundance and diversity of intestinal microorganisms. Based on the previous results, it is not clear how dietary fiber changes the relative abundance and diversity of intestinal microorganisms, with the most common explanation being the change in pH. However, it is also possible that the change in substrate of intestinal microorganisms due to the intake of feed composition and excessive secretion of intestinal protein due to intestinal inflammation may eventually change the relative abundance and diversity of intestinal microorganisms, but this needs further study.

## 5. Effects of Dietary Fiber on Immunity of Pregnant Sow and the Underlying Mechanism

### 5.1. Effect of Dietary Fiber on Immunity of Pregnant Sow

The immunity of the sow is different in different periods and plays an important role in the physiological health of the sow. After mating and entering the gestation period, the immunity and physiological metabolism of the sow undergo significant changes to ensure the correct progress of embryo implantation and embryo development as well as pregnancy completion [46]. In rat models, dietary fiber has been shown to affect serum immunoglobulins and cytokines [47]. Excessive ROS produced by over-active metabolic processes in the late pregnancy and lactation of the sow can lead to increased endotoxin levels, intestinal flora disorder, reduced SCFAs and the secretion of pro-inflammatory factors, which in turn cause local intestinal inflammation, potential damage to the intestinal microbial barrier, increased intestinal permeability, increased blood endotoxin levels, and ultimately reduce the performance of the sow and piglets [35]. Therefore, reducing the inflammatory response and ensuring normal metabolic and immune changes in the sow during the second and third trimesters and lactation are essential for the optimal performance of the sow and their offspring [46]. 

Studies have found that dietary fiber had an impact on immunity. Liu et al. found that feeding alfalfa meal dietary fiber significantly reduced ROS and endotoxin in pregnant sows, significantly reduced interleukin-6 (IL-6), lipocalin-2 and tumor necrosis factor-α (TNF-α) in serum and feces and increased interleukin-10 (IL-10) [35]. Vogt et al. found that the addition of soluble fiber to the pregnancy diet of the sow could reduce the level of the proinflammatory factor TNF-α in late pregnancy and reduce the level of maternal inflammation [48]. Shang et al. found that the intake of beet meal dietary fiber could significantly increase the levels of colostrum immunoglobulin A (IgA) and IL-10 in colostrum and IgA in the milk of lactating sows, reduce the expression of TNF-α mRNA and IL-6 in the ileum of piglets, and reduce the level of inflammation in piglets [40]. Li, Yang et al. found that feeding the sow a high content of soluble dietary fiber significantly reduced serum IL-6 and TNF-α and reduced the level of inflammation in the body [15]. 

In conclusion, the addition of feed fiber from different sources with different contents to the diet can affect the immunity of animals and change their inflammatory levels.

### 5.2. Mechanisms Underlying the Effect of Dietary Fiber on Immunity in Pregnant Sow 

Studies have shown that TNF-α can damage placental blood vessels and cause vascular embolism, which affects the nutritional supply of the fetus and the smooth progress of pregnancy. TNF-α can also cooperate with INF-γ to regulate the apoptosis of villous trophoblast cells and inhibit the development and growth of embryos [49]. The expression of inflammatory cytokines can have detrimental effects on embryonic development and function. IL-6 and endotoxin can increase the level of inflammation, which has harmful effects on the body, while IL-10 can resist the inflammation of the body through related pathways, so that the body can carry out normal activities.

After the intake of dietary fiber by the sow, gut microbes’ richness and diversity are altered, the decomposition of SCFAs is enhanced and the immune system of the sow is boosted. As a major source of metabolic energy for colonic cells, butyrate has a positive role in maintaining mucosal integrity, controlling intestinal inflammation and supporting genomic stability [50,51,52]. 

At present, it is believed that the greatest physiological effect of SCFAs is their nutritional effect on the stomach and intestinal mucosa, which can promote the growth of the digestive tract, the proliferation of intestinal epithelial cells and the enhancement of intestinal immunity [53]. Studies have shown that too little dietary fiber intake could reduce SCFAs and energy substances in the colon, resulting in insufficient energy sources for intestinal microorganisms, which in turn use mucosal glycoproteins secreted by the intestine as substrates, leading to an erosion of the mucosal barrier, increased intestinal permeability and increased systemic inflammation [54]. For example, patients with irritable bowel syndrome have abnormal intestinal flora due to an insufficient intake of SCFAs by the intestinal epithelial cells, which directly affects the distribution of tight junction proteins, resulting in a thinner intestinal flora, increased intestinal permeability and reduced protection [55]. Studies have shown that gut microbiota inhibited the NF-ƘB pathway by producing SCFAs, which led to decreased production of inflammatory cytokines and chemokines such as IL-6, and a suppression of inflammatory responses [56]. It is possible that TNF-α production is suppressed through the following mechanism. In the resting state, NF-ƘB is prevented by IƘBs from entering the nucleus to function. When the protease is activated, IƘBs is phosphorylated, and then NF-ƘB enters the nucleus, regulates the DNA of the cell nucleus and initiates transcription and expression to produce TNF-α. However, in the presence of butyric acid, it can inhibit the phosphorylation and degradation of IƘBs, thereby inhibiting the translocation of NF-ƘB and the secretion of TNF-α [57] (Figure 2). Previous studies have shown that GPR109A could be activated by the butyric acid generated in the gut, promote regulatory T cell differentiation, increase the expression of anti-inflammatory factor IL-10, and reduce the levels of inflammatory factors IL-6 and IL-17 to enhance the anti-inflammatory ability of macrophages and dendritic cells [31,58]. In colonic epithelial cells, propionate and acetate have been shown to promote intestinal and immune homeostasis through GPR43, including by protecting the integrity of intestinal epithelial cells and having anti-inflammatory effects [59,60]. The SCFA-mediated increase in glucose-derived pyruvate and acetyl-CoA levels in eukaryotic cells leads to the accumulation of citrate, its transport to the cytosol and its subsequent conversion into cytosolic acetyl-CoA by ATP- citrate lyase (ACLY) to provide energy for the body. ACLY is the key cytosolic enzyme that converts citrate to acetyl-CoA, which is needed for histone acetyltransferase (HAT)-dependent histone acetylation [61]. Notably, SCFAs are able to regulate gene expression at the epigenetic level by modulating the activity of HATs and HDACs [60]. In addition, gut microbes can use the metabolites produced by fiber decomposition to stimulate the proliferation and differentiation of B cells, thereby increasing the level of circulating IgG and enhancing the body’s immunity. This may be related to fiber breakdown products (SCFAs): 1. SCFAs can increase B cell metabolism, and SCFAs can be converted to acetyl-CoA in B cells, which increases the intracellular level of acetyl-CoA, and more acetyl-CoA enters the mitochondrial tricarboxylic acid cycle to produce more energy. Acetyl-CoA can also be used to synthesize fatty acids, and B cells can take advantage of the additional energy and material basis provided by SCFAs for better cell activation, differentiation and immunoglobulin production. 2. SCFAS regulate gene expression, so SCFAs could significantly promote the expression of Ig-related genes, such as *Xbp1* (X-box binding protein 1), *Irf4* (interferon regulatory factor 4) and *Aicda* (activation-induced cytidine deaminase), which are required for B-cell proliferation, differentiation and Ig type conversion [62] (Figure 2). 

In conclusion, fiber intake in the sow can reduce the oxidation level of the body, reduce pro-inflammatory factors, increase anti-inflammatory factors, make the animals healthier, be more beneficial to intestinal digestion and the absorption of nutrients, improve embryo implantation and development and improve growth performance and reproductive performance.

## 6. Summary

Dietary fiber has been reported to reduce stereotypic behavior and increase satiety in the sow, but the global decline in antimicrobial use has made dietary fiber important in some ways. As the seventh nutrient, dietary fiber plays a significant role in improving reproductive performance in today’s reduced-antibiotic-use environment and can also improve the composition of intestinal microorganisms without causing adverse effects on the body. In addition, dietary fiber also plays a role in improving immunity and reducing inflammation in the sow, but there are relatively few relevant studies. 

There are a few problems with the current research: 1. The optimal ratio of soluble fiber to insoluble fiber is difficult to determine, and the role of adjusting the processing methods and addition timing in improving the nutritional value of dietary fiber should be assessed to determine better feeding standards to achieve optimal reproductive performance. In addition, the specific mechanisms by which fibers regulate primordial follicle activation remain to be investigated. 2. The studies on gut microbiota have identified only a few strains with beneficial effects on the gut, and most of them remain at the genus level. 3. There are few studies on the effects of fiber on immunity. To assess the effect of fiber on inflammation, more active studies should be conducted on the regulatory mechanism pathways. In addition, there are few studies on the effects of feeding different fibers from different sources on intestinal health and inflammatory responses in sows and their offspring.

## Figures and Tables

**Figure 1 microorganisms-11-02292-f001:**
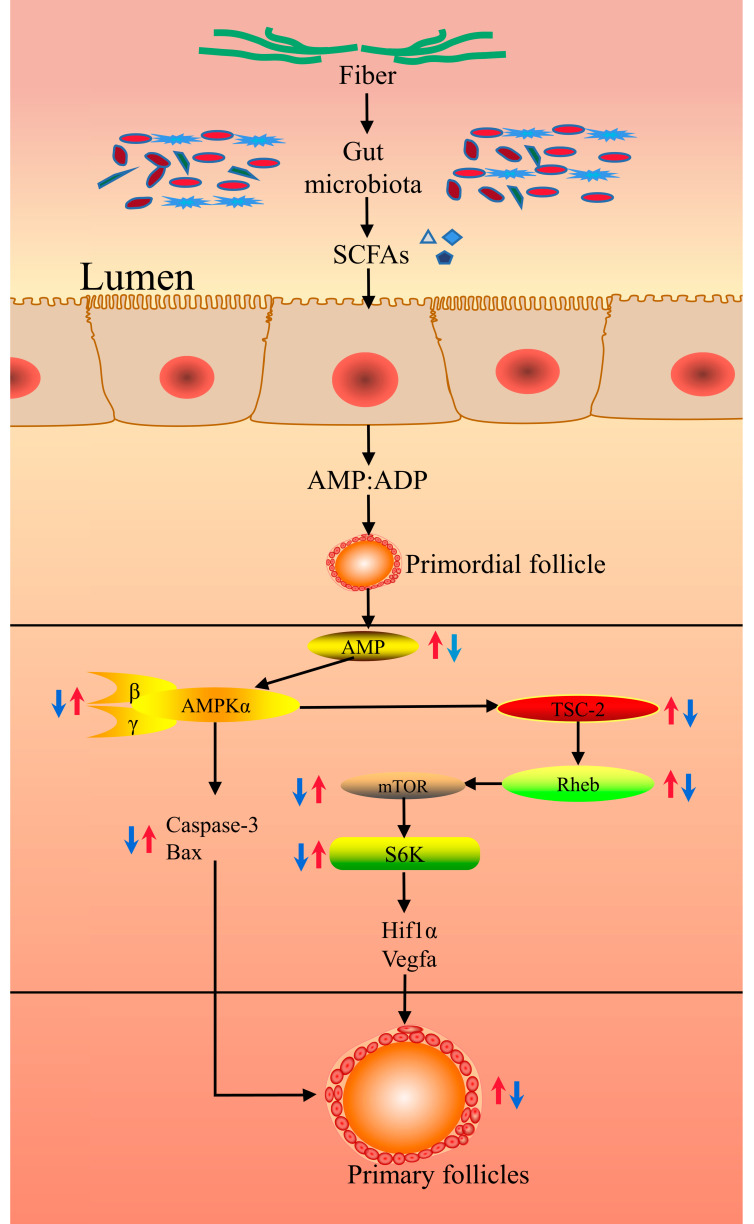
Mechanisms of signaling from the gut microbiota to the sow ovary. SCFAs = short-chain fatty acids; AMP = adenosine monophosphate; ADP = adenosine-diphosphate; AMPK = AMP-activated protein kinase; TSC-2 = tuberous sclerosis complex 2; Rheb = Ras-homolog enriched in brain; mTORC1 = mammalian target of rapamycin 1; ERK1/2 = extracellular regulated protein kinases 1/2; Caspase-3 = cysteine aspartic acid-specific protease 3; BAX = B-cell lymphoma 2 associated X protein; *Hif1α =* hypoxiainduciblefactor-1; *Vegfa =* vascular endothelial growth factor. Blue arrow: Down-regulation; Red arrow: Up-regulation.

**Figure 2 microorganisms-11-02292-f002:**
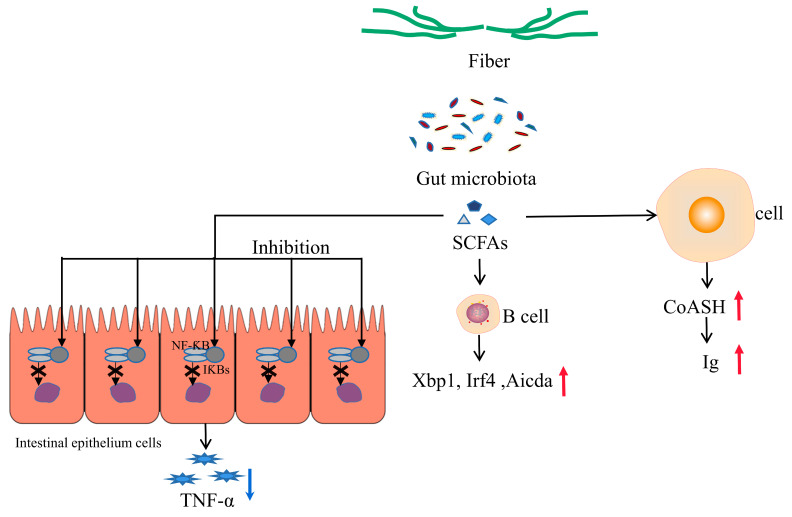
Mechanisms of signaling from the gut microbiota to the pro-inflammatory factors. SCFAs = short-chain fatty acids; NF-ƘB = nuclear factor kappa-B; IƘBs = inhibitory kappa B protein; TNF-α = tumor necrosis factor-α, CoA-SH = acetyl-CoA; Ig = immune globulin; *Xbp1* = X-box binding protein 1; *Aicda =* activation-induced cytidine deaminase; *Irf4 =* interferon regulatory factor 4. Blue arrow: Down-regulation; Red arrow: Up-regulation.

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
