# Peer review of "Effect of Dietary Fiber on Reproductive Performance, Intestinal Microorganisms and Immunity of the Sow: A Review"

_microorganisms, 2023, doi:10.3390/microorganisms11092292_

Round 1

Reviewer 1 Report

Review of the paper Effect of dietary fiber on reproductive performance, intestinal microorganism and immunity of the sow: a Review.

The manuscript describes important and current topic of dietary fiber rationale use in swine nutrition, with special emphasize on sows. Fiber perception was changing gradually during last 30 years. Previously it was considered as antinutrient for nonruminant species, however, many research showed important role of fiber for intestine function and health. Most of those research concerned human nutrition, and experiments in animals were a kind of succession. Presently, fiber is perceived as one of the main pro-health nutrient and its important role in nonruminant nutrition is appreciated, but in this condition that the level of fiber in diet is proper, and proportions among different fractions of fiber are correct. In fact, proper levels of crude fiber in individual groups of animals are pretty well recognized, but there is no valuable information about the rest of dietary fiber fractions. That is why, I was really excited with the proposition to review this manuscript, because I wondered that at least some problems will be solved thanks to it. Unfortunately, I am a little disappointed. The manuscript contains hugeness of general information, some concepts of potential physiological mechanisms, but no any detailed practical information to use in practice. It is very weak point of this manuscript, however, there are also some strong points, like deep analysis of potential physiological mechanisms of fiber action in seemingly far away processes. That is why the manuscript is very difficult to assess.     

The most important problem of the manuscript seem to be language, which is poor, and full of mistakes. The Authors write very long sentences, what generates grammar problems. Generally the manuscript is difficult to read and understand in many parts. The number of grammar mistakes is too large to point all of them in the review. In this point, the Authors should look for some advice. The next problem is many unnecessary repetitions. Probably they come from the construction of manuscript with division to sections and subsections. Sometimes the 1st subsection entitled “Effect…” many similar information that subsection 2 entitled “Mechanism…”. That is why sometimes the reader has a strong feeling that the same information was given twice.   

Next important problem is the way of interpretation of reference data. First of all is a glorification of fiber, omitting or decreasing the importance of the other nutrients. The Authors like very strong definitions like: “Dietary fiber is a key nutrient in the diet of sows” (line 209). So the question is if protein, fat or other carbohydrates are not important? The Authors seem to forget, that in nonruminant species nutrition the most important is nutrient balance, and fiber also must be balanced to more important protein, and energy sources. Reading the manuscript sometimes I had a feeling that the Authors believe the sows could be fed only with fiber. Certainly, the main topic of the manuscript is fiber, but it does not mean that the Authors should forget the other nutrients.

Sometimes the Authors try to cumulate information in one sentence, and some certain actions of fiber are mixed up with potential (line 28-30). The order of actions is also sometimes a little strange, like in line 28-30, where animal welfare or reproductive performance are forward to fecal score which is undoubtedly the first visible effect. Sometimes, the information in one sentence or in two consecutive sentences are inconsistent. In line 110-112 the Authors claim fiber supplementation did not affect colostrum yield, but increased colostrum intake. How is it possible that piglets could take more colostrum if the sow did not produce it in larger amount? In line 114-119 the information is difficult to understand. What does it mean “the number of healthy litter produced by the sow”? And why this parameter was significantly different if there was no significant differences in any of important litter parameters described in the next sentence?

The most of sections ends with a kind of conclusion, but this conclusion is sometimes poorly based on the information in section, e.g. line 124-125 “…the reproductive performance of sow can be greatly improved and enhanced by feeding dietary fiber…”. It is next example of fiber glorification, but also not every cited papers showed improvement on reproductive performance. The Authors use many general statements, but omit any specific data.  That is why it is difficult to value the information.

In my opinion the manuscript is valuable, because it raises important issue of modern swine nutrition. However, it needs substantial improvement before acceptance. In my opinion the Authors should:

1.       Describe changing situation in fiber meaning. Why 30 years ago there was no problem of fiber in diets, and now the problem is large, and fiber needs to be supplemented (the Authors should look for the information about fiber content in basal row materials like cereals).

2.       Use much more data form references, not only general statements

3.       Describe relationships between genetics and nutrition (the requirements for fiber are always the same, or maybe some genetics are more sensitive, is the reaction of every breeds for fiber supplementation always the same, or could be different, etc.)  

4.       Eliminate inconsistence in information, especially in section 2 and 3.

5.       Reduce sentences length in whole manuscript.

6.       Define most important parameters, because sometimes it is not clear what do they mean (like in line 132-134). It is not clear what does it mean “increasing number of litters, the number of live litters” and what is the difference between “the number of piglets born alive” and “the number of piglets born effective”.

7.       And very important, the Authors should look for advice of native English speaker, to improve grammar of the text and make it more clear and informative.

8.       References must be filled with the publication year (in most of references there is only vol. pages and DOI).  

I suggest major revision.         

The number of grammar mistakes is too much to analyze them separately. The Authors should look for native English speaker help to improve the text, and make it more clear and informative.

Author Response

Please find the responses in the attachment. Thank you.

Reviewer 2 Report

This review discusses dietary fiber, a substance that cannot be digested by endogenous digestive enzymes but can be digested by intestinal microorganisms' cellulolytic enzymes. In the past, fiber was considered an anti-nutrient as it hindered digestion and absorption by reducing contact between food and digestive enzymes in the small intestine. However, recent studies show that dietary fiber can provide energy for pigs through intestinal microbial fermentation and can have positive effects on sow welfare, reproductive performance, intestinal flora, and immunity. The review covers the composition and classification of dietary fiber, its impact on sow reproductive performance, intestinal microorganisms, and immune index. It offers valuable insights for the application of dietary fiber in sow production. In my opinion, it also offers a balanced view of the topic, making it a valuable piece of work. The literature used in this study has been meticulously selected to align with the topic, and all sources are up-to-date. Reading this manuscript has been an absolute delight due to its unique perspective and the meticulous preparation put into it.

Round 2

Reviewer 1 Report

The Authors made good job improving the manuscript. The most important seem to be language revision. Thanks to it the text is much easier to understand, more comprehensive and informative. In my opinion there is still some missing information, but maybe of minor importance. One of this information is fact, that during last 30 years the concentration of crude fiber in cereals decreased dramatically, probably because of breeding program directed only to yield. That is why the problem of fiber deficiency in diets appeard. Generally, the Authors focused only on animals, and in my opinion it would be better the wider look for the problem. However, taking into account the importance of the topic, I am inclined to suggest acceptance of the manuscript in present form. I suggest to include some words about fiber concentration in cereals, but it is not obligatory, rather a goodwill of Authors.
